# Lymph Node Involvement in Early-Stage Cervical Cancer: Is Lymphangiogenesis a Risk Factor? Results from the MICROCOL Study

**DOI:** 10.3390/cancers14010212

**Published:** 2022-01-02

**Authors:** Matteo Tantari, Stefano Bogliolo, Matteo Morotti, Vincent Balaya, Florent Bouttitie, Annie Buenerd, Laurent Magaud, Fabrice Lecuru, Benedetta Guani, Patrice Mathevet

**Affiliations:** 1Gynecology Department, Centre Hopital-Universitaire Vaudois, 1011 Lausanne, Switzerland; Matteo.Morotti@chuv.ch (M.M.); v.balaya@hopital-foch.com (V.B.); Benedetta.Guani@chuv.ch (B.G.); Patrice.Mathevet@chuv.ch (P.M.); 2Academic Unit of Obstetrics and Gynecology, IRCCS Ospedale Policlinico San Martino, Università degli Studi di Genova, 16128 Genoa, Italy; 3Department of Obstetrics and Gynecological Oncology, “P.O del Tigullio” Hospital-ASL4, Metropolitan Area of Genoa, 16128 Genoa, Italy; dr.bogliolo@gmail.com; 4Department of Gynecology and Obstetrics, Foch Hospital, 92150 Suresnes, France; 5Department of Biostatistics, University Hospital of Lyon, 69002 Lyon, France; florent.boutitie@chu-lyon.fr; 6Department of Pathology, Hospices Civils de Lyon HCL, 69000 Lyon, France; annie.buenerd@chu-lyon.fr; 7Clinical Research and Epidemiology Department, Hospices Civils de Lyon, 69000 Lyon, France; laurent.magaud@chu-lyon.fr; 8Faculty of Medicine, University of Lyon, Claude Bernard Lyon 1, 69007 Lyon, France; 9Faculty of Medicine, University of Paris, 75006 Paris, France; fabrice.lecuru@curie.fr; 10Breast, Gynecology and Reconstructive Surgery Unit, Curie Institute, 75005 Paris, France; 11Department of Gynecology, HFR, 1708 Fribourg, Switzerland; 12Faculty of Medicine, University of Fribourg, 1700 Fribourg, Switzerland; 13Faculty of Biology and Medicine, University of Lausanne, 1015 Lausanne, Switzerland

**Keywords:** angiogenesis, cervical cancer, lymphangiogenesis, lymph-vascular space invasion, lymph nodal metastasis

## Abstract

**Simple Summary:**

The prognosis of cervical cancer is significantly influenced by lymph node involvement. The lymphatic system is the primary way of metastasis for cervical carcinoma, and lymph-vascular space invasion (LVSI) is considered the most important risk factor for pelvic lymph node metastasis (PLNM). Previous studies have not clarified the correlation between lymphangiogenesis and an increased risk of metastasis and tumor recurrence. The evaluation and identification of several markers of lymphangiogenesis may identify patients with high risk of PLNM. Our findings suggest that the lymphatic spread does not required the proliferation of new lymphatic endothelial cells. These results emphasize the importance of pre-existing peritumoral lymphatic vessels in the metastatic process in early cervical cancer.

**Abstract:**

**Background:** In patients with cervical cancer, the presence of tumoral lymph-vascular space invasion (LVSI) is the main risk factor for pelvic lymph node metastasis (PLNM). The objective of this study was to evaluate the presence of several markers of lymphangiogenesis in early-stage cervical cancer and their correlation with PLNM and tumoral recurrence. **Materials and Methods:** Seventy-five patients with early-stage cervical carcinoma underwent sentinel lymph node (SLN) sampling in association with complete pelvic lymph node dissection. Primary tumors were stained with the following markers: Ki67, D2-40, CD31 and VEGF-C. A 3-year follow-up was performed to evaluate the disease-free survival. **Results:** Overall, 14 patients (18.6%) had PLNM. Positive LVSI was seen in 29 patients (38.6%). There was a significant correlation between LVSI evidenced by H/E staining and PLNM (*p* < 0.001). There was no correlation between high Ki67, CD31, D2-40, and VEGF-C staining with PLNM or tumor recurrence. **Conclusions:** Our data support that lymphatic spread does not require the proliferation of new lymphatic endothelial cells in early-stage cervical cancer. These results emphasize the importance of pre-existing peritumoral lymphatic vessels in the metastatic process in early cervical cancer. None of the markers of lymphangiogenesis and proliferation assessed in this study were predictive of PLNM or recurrence.

## 1. Introduction

Cervical cancer is the fourth most frequent cancer in women, with an estimated 604,000 new cases in 2020 representing 6.6% of all female tumors [1]. The presence of macroscopic LN involvement and the Sedlis criteria [2]) are considered as major prognostic factors for early-stage cervical cancer and are used to guide adjuvant therapy [2,3,4,5]. More recently, other prognostic factors for pelvic lymph node metastasis (PLNM) have been identified, such as the presence of intra-tumoral lymph-vascular emboli [6,7]. According to the new FIGO classification [8], the clinical value of micrometastasis (MIC) is the same as for the macrometastasis (MAC). However, the clinical relevance of MICs is still a matter of debate, with controversial results among the literature, especially in terms of risk of recurrence [9,10,11,12,13,14,15,16,17,18,19,20]. The clinical impact of isolated tumor cells (ITC) remained unclear, and the presence of ITCs does not change the FIGO stage.

However, it is evident that the process of lymphatic metastasis appears as a key biological step in the evolution of cervical cancer. This concept has driven the clinical implementation of the SLN biopsy concept [21,22,23].

Experimental and clinicopathological studies have indicated that the new growth and morphological changes in lymphatic vessels (lymphangiogenesis) may enhance metastasis spread by favoring the entry of tumor cells into the lymphatic network [24]. Similarly, it has been demonstrated that angiogenesis, the development of new vessels in areas of new tissue growth, is an important factor for tumor proliferation and dissemination [25].

The presence of lymphangiogenesis within primary cervical cancer has been related to an increased risk of PLNM [26]. However, to date, the clinical significance of lymphangiogenesis, its relationship with an increased risk of PLNM and correlation with prognosis, remains controversial [7,27,28,29,30,31,32]. In regard with its potential implication in cervical cancer spread, it is of paramount importance to determine whether lymphangiogenesis in the tumor could be potentially used to stratify patients for therapy or risk allocation before the LN metastasis has occurred (e.g., assessment of these markers in the conization specimen). 

In the multicentric prospective clinicopathological study named MICROCOL, we aimed to assess whether known markers of cancer cell proliferation (Ki-67), tumor angiogenesis (CD31), and lymphangiogenesis (VEGF-C and D2-40) were associated with an increased risk of PLNM, including low-volume metastasis (MIC and ITC), in a cohort of women with early-stage cervical cancer.

## 2. Materials and Methods

### 2.1. Study Design and Patients

We performed a prospective longitudinal study (named MICROCOL) in seven French centers between March 2005 and June 2007. MICROCOL was approved by the appropriate ethical committee (Comité de Protection des Personnes Lyon IV). We included adult patients enrolled in the SENTICOL study [33] with stage IA2 and IB1 cervical carcinoma based on the International Federation of Gynecology and Obstetrics (FIGO) 2009 criteria. Written informed consent was obtained from all patients before study inclusion. The exclusion criteria were a contraindication to the products used for LN mapping, a history of severe allergy, concurrent pregnancy, or magnetic resonance imaging (MRI) evidence of PLNM. All patients underwent laparoscopic pelvic LN dissection, including identification of the SLN (with a double tracer: patent blue + Nanocis ^®^), followed by bilateral pelvic lymphadenectomy. Frozen section analysis was performed only on macroscopic suspicious nodes in two centers at the surgeon’s discretion and routinely in the others.

In case of negative LN after the standard pathological examination (H/E), patients were treated with radical hysterectomy or radical trachelectomy. In the case of MAC, they underwent definitive radio-chemotherapy. In the case of MICs or ITC detected after ultrastaging (H/E + IHC), they were treated like negative LN patients. The adjuvant radio-chemotherapy treatment was added in the presence of two gynecologic oncology group (GOG) risk factors [4]. The GOG risk factors refer to a trial conducted by Delgado et al. who developed a scoring system regarding the best combination of prognostic factors for disease-free survival and recurrence; irrespective of nodal status, radiotherapy is advised in the case of early-stage cervical cancer in the presence of at least two of the following unfavorable prognostic tumor characteristics: positive LVSI, a tumor diameter of ≥40 mm and a depth of invasion at ≥1/3 or ≥15 mm [4]. 

All patients had a follow up with clinical examination every 3 months for at least 3 years. Time of recurrence was calculated from the day of surgery to the day of the recurrence diagnosis. 

### 2.2. Histopathological Examination

Sentinel and non-sentinel LNs were analyzed with H/E staining of 200-µm sections. All LNs defined as negative by H/E were submitted to the ultrastaging protocol using anti-cytokeratin antibodies (AE1-AE3 antibodies, 1:500 dilution, DAKO, Trappes, France). All sections were performed at the surgical center and analyzed by an experienced pathologist. A central review of the positive slides and 10% of the negative slides were realized by three further gynecologic pathologists. ITCs were defined as cells or masses of cells measuring ≤0.2 mm, micrometastases as tumor implants larger than 0.2 mm but ≤2 mm, and macrometastases as tumor implants >2 mm. All the primary tumor specimens were immediately fixed in 10% formalin and then embedded in paraffin. The main specimen was evaluated using at least two sagittal sections and three transversal sections, leading to a minimum of eight sections per patient. An assessment in all the sections for the presence of tumor cells in the luminal spaces lined by endothelial cells (presence of LVSI) was performed. A central review of all slides was realized.

### 2.3. Immunohistochemistry (IHC)

All cervical specimens were evaluated with the following markers: Ki67 (marker of cell proliferation, 1:50 dilution, DAKO cytomation, Glostrup, Denmark), D2-40 (lymphatic marker, 1:25 dilution, Signet Laboratories, Dedham, MA, USA), CD31 (marker of blood and lymphatic vessels, JC70A, 8 μg/mL, DAKO cytomation, Denmark), VEGF-C (a marker of vascular proliferation,1:50 dilution, ZYMED Laboratory, San Francisco, CA, USA). A standard IHC protocol was performed. Briefly, specimens were de-paraffinized, rehydrated, and stained at room temperature for 1 h using a DAKO autostainer (Dako, Trappe, France). The IHC staining was evaluated with the classic avidin-biotin peroxidase anti-peroxidase complex method. diaminobenzidine chromogen (DAB) substrate was added for 7 min. Specimens were counterstained with hematoxylin (Dako, Trappe, France) and mounted. Negative control of the IGC reaction was performed by omitting incubation with the primary antibody. As a positive control, we used an invasive breast carcinoma specimen with high micro-vascular staining. Unstained sections were prepared from formalin-fixed and paraffin-embedded radical surgical specimens or conization specimens. H/E-stained specimens were examined by the study pathologist (AB) to confirm that at least 50% of each section consisted of malignant tissue. Histopathological images can be found in Appendix A (Figure A1, Figure A2 and Figure A3).

### 2.4. Immunohistochemistry Scoring

All the markers were evaluated in five different fields at 20× magnification. For MIB-1 antibody, a nuclear staining was observed in proliferating cells. The cell proliferation index (proportion of positive tumoral cells vs. the number of positive + negative tumoral cells) was calculated. Concerning D2-40 staining, two types of staining were observed. One in endothelial cells and one in the lymphatic vessels. This leads to the evaluation of the number of positive lymphatic vascular sections (calculated in three vascular “hot spots” within the malignant tumor). In addition, cytoplasmic staining of tumoral cells was observed in some cases. The cytoplasmic staining intensity of tumoral cells was evaluated from 0 (no staining) to 2 (strong immunostaining). The scores 1 and 2 were divided into sub-score A (focal reactivity < 50%) and sub-score B (diffuse reactivity > 50%). CD31 microvessel density was quantified by counting the number of vessels plus immunoreactive endothelial cells in three vascular “hot spots” within the malignant tumor. VEGF-C staining gave cytoplasmic staining in tumoral cells. The cytoplasmic staining intensity was evaluated similarly to D2-40 cytoplasmic staining, and the staining intensity was reported from 0 (no staining) to 2 (strong immunostaining).

### 2.5. Statistical Analysis

The association with PLNM and the other risk factors was analyzed with the chi-square test with a univariate model. Fisher’s exact test was used when necessary. Continuous variables were compared by the Student’s *t*-test. *p* values less than 0.05 were considered statistically significant. Variables with *p* values lower than 0.1 by univariate analysis were entered into a multivariate logistic regression model to determine independent variables associated with PLNM. The discriminating power of risk factors associated with PLNM was calculated with a ROC curve. The Kaplan–Meier method was used to describe DFS, which was calculated as the time interval between surgery and disease recurrence. A log-rank test was applied to compare the survival in different groups of patients in stratified survival analyses. The correlation between tumor recurrence and risk factors was calculated using the Cox proportional hazards model. Statistical analysis was performed using StatPlus v.8 software (AnalystSoft Inc. 2021, Walnut, CA, USA) and XLStat Biomed software (AddInsoft Inc. 2020, New York, NY, USA).

## 3. Results

### 3.1. Clinicopathological Results

The clinicopathological characteristics of the study population are presented in Table 1. A total of 51 radical hysterectomies (76.4%) and 16 trachelectomies (12.9%) were performed. Eight patients (15.6%) were treated by radio-chemotherapy only after PLND (four patients had MAC at the frozen section analysis, and four patients refused radical surgery).

The assessment of SLN was performed in all the patients. All 75 patients had pelvic bilateral lymphadenectomy (median: 6 LNs/side [range 1–15]). SLN was bilaterally detected in 54 patients (72%) in both pelvis and in 21 patients (28%) unilaterally. Frozen section examination was performed on 124 lymph nodes in 35 patients whereas all lymph nodes (SLN and non-SLN) underwent ultrastaging examination. 

Overall, 14 patients (18.7%) had positive LNs.

Four patients (5.3%) were detected by standard examination (four MAC on SLNs). In addition, the ultrastaging detected additional metastases (MICs and ITCs) in a further 10 patients.

Concerning the non-SLN, two of the seven patients with positive non-SLN had a negative SLNs: the ultrastaging of the non-SLNs detected one MIC and one ITC in non-SLNs and added two positive-LN patients.

### 3.2. Treatment

Lymph nodal assessment was the first step and was performed by laparoscopy. 

For the 4 patients who had MAC found in SLN at frozen section, surgery was aborted, and these patients were referred to radio-chemotherapy.

All other patients were treated by radical hysterectomy or trachelectomy with additional full pelvic lymphadenectomy. In the case of low-volume metastases (MICs and ITCs) discovered at the ultrastaging, only patients presenting 2 GOG risk factors [4] received the adjuvant treatment after surgery. 

### 3.3. Risk Factors of Lymph Node Involvement

Several risk factors commonly associated [2,3,12,34] with an increased risk of PLNM in cervical cancer were assessed. Increased age or surgical approach (laparoscopy vs. laparotomy) were not associated with an increased risk of PLNM. Similarly, tumor grade, tumor size, histological type, and stromal invasion were not associated with the presence of PLNM in the univariate or in multivariate analysis (Table 2). LVSI detected by H/E staining was the only statistically significant factor associated to PLNM in univariate, and in multivariate analysis (Table 2).

In particular, LVSI was evidenced by H/E staining in 29 (38.6%) cases. In 14 node-positive patients evaluated, LVSI was detected with H/E staining in 11 cases (79%); only 3 LVSI negative cases showed PLNM (*p* = 0.0009) (Table 3). The negative predictive value (NPV) of association between LVSI detected by H/E and positive lymph node was 93% with only 3 false negatives (FN). The area under the curve (AUC) of the receiver operating characteristic (ROC) curve was 0.74 (*p* < 0.0001) (Figure 1).

We also analyzed the high density of tumoral lymphatic microvessels (LVSI stained with D2-40) and the correlation with the LN status. No statistical significance (*p* = 0.23) was noted between cases with D2-40 positive staining and the presence of PLMN (Table 3).

The correlation between PLNM and the standard markers of neoangiogenesis (CD31), lymphangiogenesis (VEGF-C and D2-40) and the proliferation marker Ki-67 was studied (Table 4). At the univariate analysis, no correlation between these markers and PLNM or low-volume LN metastasis was outlined. Concerning tumor histology, only Ki-67 staining showed a statistically significant increase in cases with adenocarcinoma histotype (*p* = 0.04). 

The correlation between the different studied markers was evaluated. We found only a correlation between reduced VEGF-C staining and reduced D2-40 tumoral staining (*p* = 0.05). Interestingly no correlation between D2-40 and Ki-67 was noted and a small percentage of Ki-67 was found in D2-40 vessels.

### 3.4. Disease-Free Survival (DFS)

Within 70 months of study period, a minimal follow up of 36 months was reached.

Five (6.6%) of 75 patients had tumor recurrence. The disease-free survival (DFS) for the whole cohort was 94%. The median time to recurrence was 33 months (range, 1–36). In the case of negative LN (N0) (61 patients), we had three recurrences (4.9%), whereas two recurrences (14%) were reported in the case of PLNM. The recurrence rate was not significantly different (*p* = 0.32). In particular, we had one recurrence in a patient with one ITC in the non-SLN and one recurrence in a patient with one MIC and one ITC in the non-SLN (Figure 2).

In the Kaplan–Meier analysis, none of the following risk factors were significant: type of LN involvement (MAC, MIC, ITC), histological tumor type, tumoral diameter, LVSI identified by H/E or D2-40, VEGF-C, Ki-67 or CD-31, and the age of the patients correlated with lower survival (Table 5). In the multivariate Cox regression analysis, none of the risk factors (size, stromal invasion, grade, type of tumor, and age of patients) was associated with an increased risk of recurrence (Table 5).

## 4. Discussion

Like other malignant tumors, cervical carcinomas can promote their metastatic potential through the co-option and/or formation of blood and lymphatic vessels. Indeed, the lymphatic system is the primary way of metastasis for cervical carcinoma, and PLNM is considered the most important prognostic factor of early cervical cancer [35]. However, the correlation of lymphangiogenesis in primary tumors with an increased risk of metastasis and tumor recurrence has been a matter of debate [24].

In this prospective clinicopathological study, we found that standard markers of neoangiogenesis (CD31) and lymphangiogenesis (VEGF-C and D2-40) did not correlate with an increased risk of PLNM (including MICs and ITCs) or early recurrence in a cohort of early-stage cervical cancer patients. In our study, the presence of LVSI detected with H/E staining was the only factor associated with PLNM in the univariate and multivariate analysis. 

This study does not confirm previous findings, where CD31, VEGF-C, and D2-40 have been shown to increase the sensitivity in detecting PLNM with respect to the standard identification of LVSI by H/E. The different cohorts enrolled (patients with a mix of early or locally advanced cervical cancer), different antibodies used, and the number of patients could potentially explain the different results between the different studies.

The concept of lymphangiogenesis and its correlation with increased LN metastasis has been relatively well established in colorectal carcinoma, where a high density of tumoral lymphatic microvessels (stained with D2-40) had prognostic significance correlated with metastasis to regional lymph nodes and the liver [36]. In breast cancer, despite LN metastasis being a key prognostic factor for the survival of these patients, the concept of lymphangiogenesis has been an area of controversy, with some studies showing correlation and others showing no correlation between lymphangiogenesis regional LN metastasis [37]. In addition, lymphangiogenesis is not present or detected in some tumors, such as pancreatic ductal carcinoma, even though the spread of tumor cells to the regional LN can occur [38].

Thus, the results of this clinicopathological study seem to suggest that pre-existing lymphatic vessels (LVSI positive at H/E staining), rather than new vessels generated through lymphangiogenesis (D2-40, CD31), are associated with PLNM in early-stage cervical cancer patients. Moreover, these new lymphatics showed no evidence of high proliferation, as assessed by the lack of co-expression between D2-40 and Ki67 antigen, despite the active proliferation of carcinoma cells within the immediate vicinity.

With the assumption that most cervical cancers are caused by HPV infection, in the last few years, is emerging the role of new biological parameters, such as the circulating HPV DNA (HPV ctDNA) that could be integrated in the diagnostic and therapeutic process of these diseases. The clinical value of HPV ctDNA had already been studied in patients with metastatic anal carcinoma to assess the response to chemotherapy or immunotherapy [39,40]. 

In the field of cervical cancer, a trial conducted by Jeannot et al. showed that the complete clearance of HPV ctDNA observed by the end of treatment of chemoradiation was significantly associated with a longer DFS and could be used as a reliable marker to predict the risk of relapse [41]. Unfortunately, in this trial we did not study the HPV ctDNA; further studies will be needed to assess the impact of HPV ctDNA in early-stage cervical cancer.

Despite the short follow up (3 years only), we did not find any statistical difference in DFS in patients with or without LN involvement. Still, all patients with MAC were directly treated by radio-chemotherapy. On the other hand, patients with low-volume metastasis without any other risk factor (according to the GOG [3] and Seidlis [2] criteria) that did not receive adjuvant radio-chemotherapy did not have their DFS impacted. These results preliminary confirmed our previous findings that radio-chemotherapy does not improve DFS in early-stage cervical cancer patients in patients with low-volume metastasis. 

However, the limits of the study, the small cohort of patients and the short follow up do not allow us to provide definitive results on this subject. Further studies with a larger cohort and longer follow up are needed to confirm the clinical value of lymphangiogenesis in primary tumors and the increased risk of metastasis and tumor recurrence. 

## 5. Conclusions

In conclusion, our data show that early-stage cervical cancer does not require the proliferation of new lymphatic endothelial cells for lymphatic spread. These results emphasize the importance of pre-existing peritumoral lymphatic vessels in the metastatic process in early cervical cancer.

Future studies are required to understand other biological mechanisms driving PLNM, such as the interplay of pre-metastatic clones in the immune environment within the primary tumoral lymphatic vessels.

## Figures and Tables

**Figure 1 cancers-14-00212-f001:**
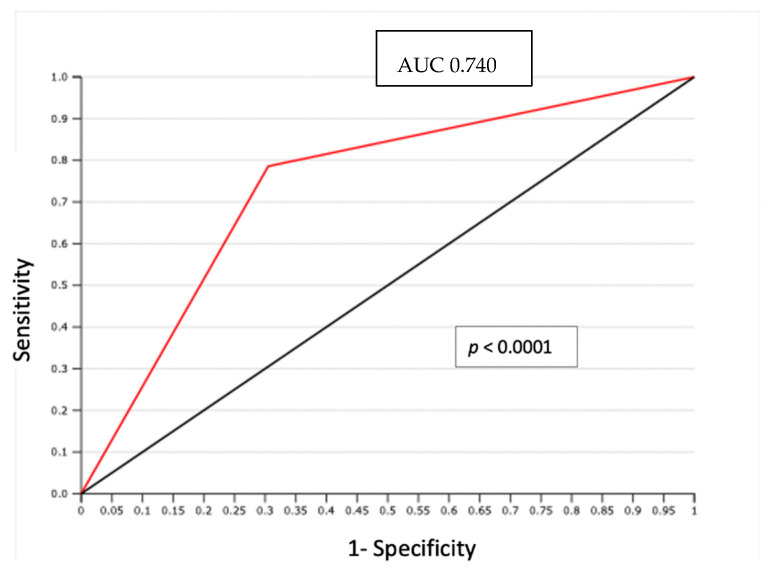
ROC curve: LVSI detected by H/S and positive lymph node.

**Figure 2 cancers-14-00212-f002:**
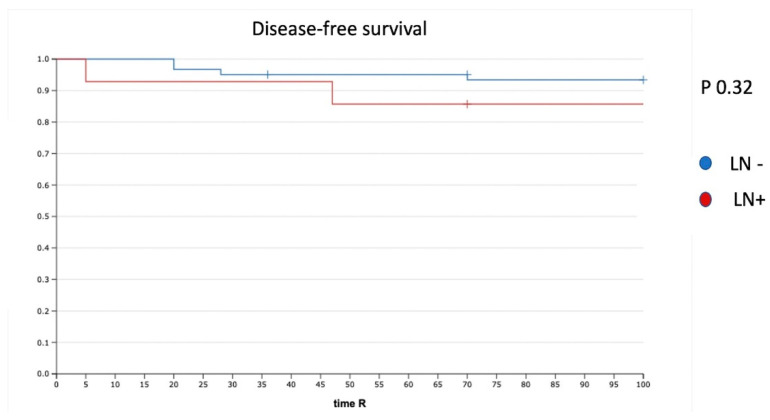
DFS in patients with positive and negative LN. LN−: negative lymph node; LN+: positive lymph node.

**Table 1 cancers-14-00212-t001:** Patients’ characteristics.

Characteristics	Patient N.		%
Age	75	Median 42.3 Range 23–85	
**Surgery type**			
LPS	63		84
LPT	12		16
**Stage**			
IA2	4		5.3
IB1	71		94.7
**Histological type**			
Adenocarcinoma	19		25.3
Squamous	54		72
Adenosquamous	2		2.7
**Grade**			
G1	23		30.6
G2	28		37.4
G3	16		21.4
n.d.	8		10.6
**PNI**			
pos	1		1.3
neg	68		90.7
n.d.	6		8

LPS: laparoscopy; LPT: laparotomy; PNI: perineural invasion; StD: standard deviation.

**Table 2 cancers-14-00212-t002:** Risk factors associated with PLNM.

Factors	N. with PLNM N. Tot (%)	Association with PLNM (*p*-Value U)	Association with PLNM (*p*-Value M)
**Age**		0.18	0.10
<50	10/54 (18.5%)		
50–70	2/16 (12.5%)		
>70	2/5 (40%)		
**Surgical approach**		0.73	
LPS	10/63 (16.6%)		
LPT	4/12 (33.3%)		
**Grade**		0.84	
G1	4/23 (17.4%)		
G2	5/28 (17.8%)		
G3	4/16 (25%)		
n.d.	1/8 (12.5%)		
**Size**		0.36	
2 cm	4/36 (11.1%)		
>2 cm	8/32 (25%)		
n.d.	2/7 (28.6%)		
**Type**		0.65	
Squamous	11/54 (20.3%)		
Adenocarcinoma	3/19 (15.8%)		
Adenosquamous	0/2 (0%)		
**Stromal invasion**		0.11	0.96
<3 mm	0/3 (0%)		
3–8 mm	2/14 (14.3%)		
>8 mm	7/23 (30.4%)		
n.d.	5/35 (14.3%)		
**LVSI (H/E staining)**		0.0009	0.03
Positive	11/29 (38%)		
Negative	3/44 (7%)		

N: number; U: univariate analysis; M: multivariate analysis; PLNM: pelvic lymph nodal metastasis; LPS: laparoscopy; LPT: laparotomy; LVSI: lymph vascular space invasion; H/E: hematoxylin-eosin.

**Table 3 cancers-14-00212-t003:** Analysis between tumor emboli and lymph nodes status in H/E and D2-40 antibody staining.

	N+	N−	Total	*p* Value (U)
**H/E Staining**				
LVSI+	11	18	29	
LVSI−	3	41	44	0.0009
**D2-40**				
LVSI+	8	26	34	
LVSI−	5	34	39	0.23

H/E: hematoxylin-eosin; LVSI+: presence of lymph-vascular space invasion; LSVI−: absence of lymph-vascular space invasion; U: univariate analysis.

**Table 4 cancers-14-00212-t004:** Correlation between proliferation markers and lymph node involvement.

Markers					*p* Value
**Ki 67**	Mean ± st.d.	Median	N.	n.d.	0.45
LN+	63.38 ± 19.74	69 (34–95)	13	1	
LN−	67.89 ± 17.3	64 (15–96)	59	2	
**CD31**	Mean ± st.d.	Median	N.	n.d.	0.28
LN+	13.32 ± 10.19	11.3 (3.33–45)	14	2	
LN−	11.34 ± 4.83	10.6 (3.75–24.4)	59	0	
**D2-40 Ab**	0	1	2	n.d.	0.43
LN+	10	1	2	1	
LN−	34	7	18	2	
**VEGF-C**	0	1	2–3	n.d.	0.31
LN+	5	5	4	0	
LN−	27	23	7	4	

LN+: positive lymph nodes; LN−: negative lymph nodes.

**Table 5 cancers-14-00212-t005:** Recurrence and risk factors.

Factors	Number of Recurrences	Kaplan–Meier *p* Value	Cox Regression *p* Value
**LN status**		0.25	
MAC	0/4 (0%)		
MIC	1/5 (20%)		
ITC	1/5 (20%)		
N0	3/61 (5%)		
**Tumor type**		0.63	
Squamous	3/56 (5.3%)		
Adenocarcinoma	2/17 (11.7%)		
Adenosquamous	0/2 (0%)		
**Age**		0.07	0.91
<50	1/54 (1.8%)		
>50	3/21 (14.2%)		
**Tumoral diameter**		1	
<2 cm	2/36 (5.5%)		
>2 cm	3/32 (9.3%)		
**LVSI (H/E)**		0.17	0.42
Negative	4/30 (13.3%)		
Positive	1/45 (2.2%)		
**LVSI (D2-40)**		1	
Negative	4/39 (10.2%)		
Positive	1/34 (2.9%)		
**VEGF-C**		1	
0	3/32 (9.3%)		
1	1/28 (3.5%)		
2–3	1/11 (9%)		
**KI-67**	Mean value	0.64	
If recurrence	72.04 ± 9.33		
If not recurrence	63.61 ± 14.99		
**CD-31**	Mean value	0.2	
If recurrence	17.10 ± 3.87		
If not recurrence	11.20 ± 9.37		

LN: lymph node; MAC: macrometastasis; MIC: micrometastasis; ITC: isolated tumoral cells; N0: negative lymph node; LSVI: lymph vascular space invasion.

## Data Availability

The data presented in this study are available on request from the corresponding author. The data are not publicly available due to restrictions of the Ethics Committee.

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
