# Peer review of "Lymph Node Involvement in Early-Stage Cervical Cancer: Is Lymphangiogenesis a Risk Factor? Results from the MICROCOL Study"

_cancers, 2022, doi:10.3390/cancers14010212_

Round 1

Reviewer 1 Report

The paper by Tantary et al., entitled « Lymph node involvement in early-stage cervical cancer : is the lymphangiogenesis a risk factor ? Result of MICROCOL study » reports the comparison between the density of vascular markers in tumor tissue, pelvic lymph node status and disease outcome in a prospective series of 75 patients with early-stage cervical cancer. No evidence of correlation between lymphatic density and pelvic lymph node status or disease outcome was observed. In contrast, a significant positive link between the presence of lymph vascular space invasion (LVSI) and pelvic lymph node status was observed.  The recurrence rate was 4.9% in node negative patients and 14% in node positive.

As a general comment, the study looks well performed. The positive link observed between LVSI and pelvic node status is a surrogate of the validity of this series of early-stage cervical cancer cases and of the statistical analysis. Since the study was designed to address the question of the value of lymphangiogenesis as a risk factor for tumor spread, and since the result is unambiguously negative, a more factual title such as “No evidence of lymphangiogenesis as a significant risk factor for tumor spread in early-stage cervical cancer” would be justified and more attractive for the reader. Obviously, this result can be discussed, but the in the field of cervical cancers, new biological parameters such as the dynamic of circulating HPV DNA have presently more potential as risk factors of relapse (Veyer et al., Int J Cancer 2019, Jeannot et al. CCR 2021) than the vascular density which is difficult to assess and cannot be repeated over time. A brief comment could be added in the discussion.

Minor points:

The number of cases is not provided in the abstract.

Some stylistic corrections are to be made. Ex

line 35:does not require ?

end of line 81 “in the”?

Beginning of line 84 In the multivariate analysis?

Line 186 …non-SLN) were submitted…

Line 200 GOG risk factor : is it possible to explain ?

Line 209 was the ?

Line 244 occurred ?

Author Response

Dear Reviewer,

We appreciate the time and effort that you have dedicated to providing your valuable feedback on our manuscript. We are grateful for your insightful comments on our paper. We have been able to incorporate changes to reflect most of your suggestions. We have highlighted the changes within the manuscript.

  • Comment from Reviewer 1 “in the field of cervical cancers, new biological parameters such as the dynamic of circulating HPV DNA have presently more potential as risk factors of relapse (Veyer et al., Int J Cancer 2019, Jeannot et al. CCR 2021) than the vascular density which is difficult to assess and cannot be repeated over time. A brief comment could be added in the discussion.”

Response: Thank you for pointing this out. We agree with this comment. Therefore, we have mentioned it in the discussion:

“With the assumption that most cervical cancers are caused by HPV infection, in the last years, is emerging the role of new biological parameters, such as the circulating HPV DNA (HPV ctDNA) that could be integrated in the diagnostic and therapeutic process of these diseases. The clinical value of HPV ctDNA had already been studied in patients with metastatic anal carcinoma to assess the response to chemotherapy or immunotherapy. 

In the field of cervical cancer, a trial conducted by Jeannot et al. showed that the complete clearance of HPV ctDNA observed by the end of treatment of chemoradiation was significantly associated with a longer DFS and could be used as a reliable marker to predict the risk of relapse“.

[page 9,paragraph 13, line 303]

  • Comment from Reviewer 1: Since the study was designed to address the question of the value of lymphangiogenesis as a risk factor for tumor spread, and since the result is unambiguously negative, a more factual title such as “No evidence of lymphangiogenesis as a significant risk factor for tumor spread in early-stage cervical cancer” would be justified and more attractive for the reader.

Response: thank you for your suggestion. I'm sorry, but, after consulting the other editors, we decided not to change the title which was the subject of debate when the manuscript was written, and which finally obtained the agreement of all the co-authors in the present form.

  • Comment from Reviewer 1: does not require ?

Response: thanks for the correction, it has been modified according to your instructions. [page 1,paragraph 1, line 36]

  • Comment from Reviewer 1: The number of cases is not provided in the abstract.

Response: We have, accordingly, modified the abstract mentioning the number of cases.

[page 1,paragraph 2, line 42]

  • Comment from Reviewer 1: “in the”?

Response: in the tumor

[page 2,paragraph 3, line 82]

  • Comment from Reviewer 1: GOG risk factor : is it possible to explain ? (line 200)

Response: We agree with your suggestion and we have incorporate the explanation of the GOG criteria following the resutls emerging from the trial conducted by Delgado et al..

We decided to move the explanation of these risk factors to the section on materials and methods where the criteria were first mentioned.

[page 3,paragraph 2, line 110]

  • Comment from Reviewer 1: …non-SLN) were submitted…

Response: We have accordingly modified to clarify that all LN (SNL and non-SLN) underwent ultrastaging examination.

[page 5,paragraph 9, line 187]        

  • Comment from Reviewer 1: was the ?

Response: LVSI was the only risk fact that shows a significatn association with PLNM

[page 6,paragraph 11, line 216]

  • Comment from Reviewer 1: occurred?

Response: Thank you for your correction, we deleted occurred.

[page 8,paragraph 12, line 253]

Sincerely,

Benedetta Guani and Matteo Tantari

Reviewer 2 Report

This work is extremely interesting as it accurately assesses the role of lymphangiogenesis as a risk factor for cervicocarcinoma in the early stages. the authors also highlight the absence of correlation between the expression of ki67, VEGF-C and PLNM. The discovery of this lack of correlation could help to identify new anticancer agents, interfering with molecular targets expressed by the tumor's microenvironment or by the tumor cell itself. The authors should therefore mention the new therapeutic perspectives that are emerging in the management of metastatic cancer of the uterine cervix such as for example "Targeted Agents in Cervical Cancer: Beyond Bevacizumab" described by Marquina G, Manzano A, Casado A. "Cervical cancer: are there potential new targets? An update on preclinical and clinical results"described by Tomao F.et al In the discussion, the authors should better highlight the limits of the study and in particular the sample size

Author Response

Dear Reviewer,

We appreciate the time and effort that you have dedicated to providing your valuable feedback on our manuscript. We are grateful for your insightful comments on our paper. We have been able to incorporate changes to reflect most of your suggestions. We have highlighted the changes within the manuscript.

-           Comments from Reviewer 2 “The authors should therefore mention the new therapeutic perspectives that are emerging in the management of metastatic cancer of the uterine cervix such as for example "Targeted Agents in Cervical Cancer: Beyond Bevacizumab" described by Marquina G, Manzano A, Casado A. "Cervical cancer: are there potential new targets? An update on preclinical and clinical results"described by Tomao F.et al.”

Response: Thank you for this suggestion. It would have been interesting to explore this aspect. However, in the case of our study, it seems slightly out of scope because in our manuscript we discussed and treated cervical cancer in its early stages with the possible prognostic correlation related to neo-lymphangiogenesis. The role of lymphangiogenesis in patients with metastatic cancer was not considered in our study, in which case a different population should be included.

  • Comment from Reviewer 2: “In the discussion, the authors should better highlight the limits of the study and in particular the sample size”

Response: We have, accordingly add to our discussion the limits of our study.

[page 9,paragraph 13, line 319]

Sincerely,

Benedetta Guani and Matteo Tantari
